# Novel Brassinosteroid Analogues with 3,6 Dioxo Function, 24-Nor-22(*S*)-Hydroxy Side Chain and *p*-Substituted Benzoate Function at C-23—Synthesis and Evaluation of Plant Growth Effects

**DOI:** 10.3390/ijms25147515

**Published:** 2024-07-09

**Authors:** Sebastián Jorquera, Mauricio Soto, Katy Díaz, María Nuñez, Mauricio A. Cuellar, Andrés F. Olea, Luis Espinoza-Catalán

**Affiliations:** 1Departamento de Química, Universidad Técnica Federico Santa María, Avenida España 1680, Valparaíso 2340000, Chile; sebastian.jorquera@sansano.usm.cl (S.J.); mauricio.sotoc@usm.cl (M.S.); katy.diaz@usm.cl (K.D.); maria.nunezg@sansano.usm.cl (M.N.); 2Facultad de Farmacia, Escuela de Química y Farmacia, Universidad de Valparaíso, Av. Gran Bretaña 1093, Valparaíso 2340000, Chile; mauricio.cuellar@uv.cl; 3Centro de Investigación, Desarrollo e Innovación de Productos Bioactivos (CINBIO), Universidad de Valparaíso, Valparaíso 2340000, Chile; 4Grupo QBAB, Instituto de Ciencias Químicas Aplicadas, Facultad de Ingeniería, Universidad Autónoma de Chile, El Llano Subercaseaux 2801, Santiago 8900000, Chile

**Keywords:** brassinosteroid analogs, synthesis, plant growth, molecular docking

## Abstract

Brassinosteroids (BRs) are an important group of polyhydroxylated naturally occurring steroidal phytohormones found in the plant kingdom in extremely low amounts. Due to the low concentrations in which these compounds are found, much effort has been dedicated to synthesizing these compounds or their structural analogs using natural and abundant sterols. In this work, we report the synthesis of new brassinosteroid analogs obtained from hyodeoxycholic acid, with a 3,6 dioxo function, 24-Nor-22(*S*)-hydroxy side chain and *p*-substituted benzoate function at C-23. The plant growth activities of these compounds were evaluated by two different bioassays: rice lamina inclination test (RLIT) and BSI. The results show that BRs’ analog with *p*-Br (compound **41f**) in the aromatic ring was the most active at 1 × 10^−8^ M in the RLIT and BSI assays. These results are discussed in terms of the chemical structure and nature of benzoate substituents at the *para* position. Electron-withdrawing and size effects seems to be the most important factor in determining activities in the RLIT assay. These results could be useful to propose a new structural requirement for bioactivity in brassinosteroid analogs.

## 1. Introduction

Brassinosteroids (BRs) are natural polyhydroxysteroid compounds that have been recognized as a different class of plant hormones [1,2]. BRs were first described in the 1970s when Mitchell et al. reported the promotion of stem elongation and cell division by treatment with organic extracts of rapeseed (*Brassica napus*) pollen [3]. In 1979, brassinolide (**1**) (Figure 1) was isolated from *Brassica napus* pollen [3,4], and it has been recognized as the most active BR out of 70 natural occurring BRs isolated so far [5]. In Figure 1, the chemical structures of the most common and active natural BRs are shown, namely, brassinolide, 24-epibrasinolide (**2**), castasterone (**3**) and 24-epicastasterone (**4**). It is currently known that BRs regulate plant growth and development by producing an array of physiological changes and eliciting very important functions, such as plant growth regulation and cell division and differentiation in young tissues of growing plants [6,7,8,9]. BRs also play an important role in molecular and physiological responses in plant growth and abiotic stress such as drought, salinity, high temperature, low temperature, and heavy metal stresses [7,8,9].

These compounds are found at extremely low concentrations in nature, and therefore much effort has been dedicated to synthesizing these compounds or their structural analogs using natural and accessible sterols [1]. In this task, side chain design remains the main synthetic challenge [10]. In that direction, an efficient method for construction of 23-arylbrassinosteroidal side chains was described via Heck coupling and asymmetric dihydroxylation as key steps, and several (22*R*,23*R*)-23-arylbrassinosteroids (compounds **5**–**10**, Figure 1) were synthesized [11]. Biological activity of these compounds was evaluated using the rice lamina inclination test (RLIT) bioassay, and analog **5** was as active as 24-epibrassinolide [12].

A series of 23-*p*-substituted phenyl BRs analogs (compounds **11**–**20**, Figure 1) was designed by molecular docking and synthesized using the Horner–Wadsworth––Emmons (HWE) reaction [13]. These analogs were tested in several biological assays, in which compounds **11** and **12** were the most active [13]. The series **21**–**31** comprises phenyl brassinosteroid analogs that were prepared via alkene cross-metathesis, and all of them were docked into the active site of BRI1 using AutoDock Vina. Plant growth-promoting activity was measured using the bean second internode (BSI) and Arabidopsis root sensitivity assays and compared with **2** [14]. The results indicate that phenyl substitution with a F atom at *ortho* or Cl atom at *meta* position generates the most active compounds of this series (**21** and **24**), whereas substitution with methyl or trifluoromethyl groups causes a significant decrease in or complete loss of plant activity [14].

In the same context, our research group reported the synthesis and biological evaluation of 24-norcholane type BR analogs, with 22(*S*)-hydroxy and benzoate functions at the C-23 position of the side chain (compounds **32**–**35**, Figure 2) [15,16,17]. Analog **32** exhibited similar activity behavior to **1**, which was used as a positive control in the RLIT, at all tested concentrations [18]. Additionally, the synthesis and biological activity of BR analogs with benzoate groups at the C-22 position, substituted with a F atom at the “*ortho*” or “*para*” positions, (**37** and **38**, Figure 2) were described [19]. Plant growth-promoting activities were evaluated by using RLIT and BSI biotests and the obtained results indicate that compound **38** (F atom in *para* position) is the most active BR analog, and in some cases was even more active than **1**. Molecular docking results show that analogs with a fluorine-substituted aromatic ring in the side chain adopt an orientation that is similar to that predicted for brassinolide [19].

Considering the important effect on biological activity produced by incorporating substituted aromatic rings in the side chain, herein the synthesis of 24-norcholane type BR analogs with 3,6-dioxo function and different substituents at the “*para*” position of the aromatic ring (compounds **41a**–**f**, Figure 2) is described.

The idea of attaching electron-donating and electron-withdrawing substituents to the aromatic ring is to evaluate the effect of electron density and substituents size on biological activity, as reported by other authors [13,14]. All obtained compounds were characterized by IR, HRMS, and 1D and 2D NMR spectroscopic techniques. Additionally, these compounds were evaluated for hormonal activity by plant bioassays RLIT and BSI. The obtained results could be important to propose a new structural requirement for BR analog bioactivity.

## 2. Results and Discussion

### 2.1. Chemical Synthesis

Commercially available hyodeoxycholic acid (**39**, Figure 2) (AK Scientific Inc., Union City, CA, USA) was used as starting material for preparation of new BR analogs, **41a**–**f**. The synthesis route developed is shown in Figure 3.

In a first step, full Jones oxidation of hyodeoxycholic acid gives dioxidized compound **42** with 97.0% yield, as reported for other bile acids [20]. The presence of both ketone groups is evidenced by ^13^C NMR spectrum (Appendix A), where the signals observed at δ_C_ = 210.92 and 208.76 ppm are assigned to carbons C-6 and C-3, respectively. Both signals were correlated and identified by combined 1D and 2D NMR experiments (Appendix A). Oxidative decarboxylation of the side chain in compound **42**, with the Pb(OAc)_4_/Cu(OAc)_2_ system and following a reported procedure [15,21,22,23], leads to olefin **43** with 42.7% yield. The formation of a terminal double bond at the C-22 position on the side chain was confirmed by ^1^H-NMR and ^13^C-NMR spectra. Namely, signals appearing at δ_H_ = 5.65 ppm (1H, ddd, *J* = 17.2, 10.1 and 8.5 Hz); 4.91 ppm (1H, dd, *J* = 17.2 and 1.9 Hz) and 4.83 ppm (1H, dd, *J* = 10.1 and 1.9 Hz) were assigned to H-22, H*_trans_*-23 and H*_cis_*-23, respectively. In the ^13^C NMR spectrum, signals observed at δ_C_ = 144.69 and 111.96 ppm were assigned to C-22 and C-23, respectively. The presence of a terminal double bond was confirmed by ^13^C DEPT-135, 2D HSQC and 2D HMBC NMR experiments (Appendix A). All NMR spectroscopic data registered for compound **43** were consistent with that reported [24].

Isomerization from 5β to 5α on compounds **43** and **44** was performed under acidic conditions (HCl 2.5%, CH_3_OH, reflux) [25,26,27]). Isomerization was verified by analyzing changes in the chemical shift and shape of the H-5 signal, and the chemical shift of the C-19 signal, in the ^1^H and ^13^C NMR spectra, respectively. These data are summarized in Table 1.

Additionally, registered IR and NMR spectroscopic data (Appendix A) were consistent with those reported [26].

Upjohn dihydroxylation (OsO_4_/NMMO/(CH_3_)_2_CO) of alkene **44** produces an epimeric mixture of **40** and **40a** (Scheme 1) with 64.3% yield. The relative amount of epimers in this mixture was estimated as **40**:**40a** = 4:1 ratio, by using relative integration areas of CH**_3_**-21 group signals for both compounds in the ^1^H NMR spectrum. The signal observed at δ_H_ = 0.962 ppm (d, *J* = 6.6 Hz, 3H) was assigned to CH_3_-21 in epimer **40** (C-22*S*), while the signal observed at δ_H_ = 0.923 ppm (d, *J* = 6.5 Hz, 3H) was assigned to CH_3_-21 in epimer **40a** (C-22*R*). The corresponding integral values were 6.00 and 0.74, respectively, which give the relative integration ratio 6.00 (**40**):0.74 (**40a**) = 4:1 (Appendix A). The observed stereoselectivity of this reaction is in line with that reported for the dihydroxylation of terminal alkenes in steroids with similar structures [15,16,17,23,24,27,28]. Additionally, the presence of glycol function in C22–C23 and full structure was confirmed by ^13^C, ^13^C DEPT-135, 2D HSQC and 2D HMBC NMR experiments (Appendix A).

Selective benzoylation of primary alcohol at C-23 of epimeric mixture **40**/**40a** with the corresponding benzoyl chlorides, according to a reported protocol [15,16,17], followed by CC of reaction products, and subsequent crystallization purification, gives compounds **41a**–**f** with 25.9%; 8.4%; 30.6%; 23.3%; 19.9% and 14.9% yields, respectively (Scheme 1). Since the mixture was benzoylated, in none of these cases was it possible to isolate the benzoylated epimers 22(*R*). Presumably, this is due to epimer **40a**, which is a very minor product, being lost in the purification process. However, pure glycol **40** was obtained with 36.2% yield from saponification reaction of benzoate **41a** with the K_2_CO_3_/CH_3_OH system in reflux, according to the described procedure [16]. Finally, compounds **40** and **41a**-**f**, were fully characterized by IR, 1D, 2D NMR and HRMS spectroscopic techniques (Appendix A).

### 2.2. Biological Activity

#### 2.2.1. Bioactivity in the Rice Lamina Inclination Test (RLIT)

Bioactivity in all compounds was evaluated by using RLIT, a highly sensitive and specific assay for BRs, and one of the most widely used methods worldwide. Briefly, the bending angles between the leaf and sheath of rice-growing plants are measured in the presence of exogenously applied BR analogs. The results obtained at different concentrations of BR analogs and brassinolide, used as positive control, are given in Table 2.

The data in Table 2 show that benzoylation of primary alcohol at C-23 brings about significant changes in activity. These changes depend on the nature of *para* position substituent and the tested concentration. For example, at the lowest tested concentration, analogs with non-substituted benzoate (**41a**) or *p*-CH_3_-substituted (**41b**) show activities that are slightly lower than that observed for **40**, whereas the analog with *p*-OCH_3_ is completely inactive (**41c**). On the other hand, the analogs carrying a benzoate group substituted with halogen atoms exhibit enhanced activities as compared to **40**. From all these analogs, **41f** (*p*-Br) is the most active, and its activities, at all tested concentrations, are higher than that observed for **40**. Moreover, at 1 × 10^−8^ M, the activity of **41f** is comparable to that observed for brassinolide (see also Appendix A). Thus, considering the nature of the substituent at the *para* position, the order of activities is **41f** (*p*-Br) > **41d** (*p*-F) = **41e** (*p*-Cl) > **41a** (*p*-H) = **41b** (*p*-CH_3_) > **41c** (*p*-OCH_3_).

#### 2.2.2. Bioactivity in Bean Second Internode Bioassay

All compounds were tested using the BSI bioassay as well, which has also been used to detect biological activity in phytohormones [29,30]. All BR analogs were applied at 1 × 10^−8^ M concentration according to a previously described procedure [19]. The values obtained from the bean second internode elongation test are summarized in Table 3, and Appendix A.

From the data shown in Table 3, it can be observed that the analog with no benzoate function, **40**, is more active than all the analogs with a benzoate group, except **41f**. The activity of analogs in this bioassay can be arranged in descending order of biological activity: **41f** (*p*-Br) > **41c** (*p*-OCH_3_) ≈ **41e** (*p*-Cl) > **41d** (*p*-F) > **41a** (*p*-H) > **41b** (*p*-CH_3_).

Considering that the only structural difference between these analogs is in the nature of substituents at the *para* position, these results should be explained in terms of induction and/or steric effects that are not present in the unsubstituted analog **41a** (*p*-H). In this context, it is known that halogen atoms exhibit similar electron-withdrawing inductive effects, whereas methyl and methoxy groups are electron-donating groups. On one hand, the results of the RLIT (Table 2) suggest that the activity of these molecules is enhanced by decreasing electron density on the aromatic ring. However, at the lowest tested concentration, the bending angle measured in the presence of **41f** (*p*-Br) is almost double those measured for **41e** (*p*-Cl) and **41d** (*p*-F). Therefore, the *p*-Br atom effect on this activity may be attributed to both inductive and size effects of this atom. On the other hand, measuring the activity of BR analogs using the BSI bioassay indicate that substitution in the *para* position leads to completely different results. For example, analogs **41c** (*p*-OCH_3_) and **41e** (*p*-Cl), which are substituted with electron-donating and electron-withdrawing groups, respectively, exhibit almost the same activity. This result suggests that inductive effects play no role in the bioactive property determined by this assay.

Interestingly, the reported activities of analogs **37** and **38**, which have a F atom-substituted benzoate group, are similar to those measured for **1** and at least twice those found for **41d** [19]. The main structural difference between these analogs and **41d** is in the presence of hydroxyl groups at C-2 and C-3 instead of the ketone group at C-3. Thus, these results indicate that the effect of the electronic density and size of the alkyl chain on biological activities depends on the entire BR structure, and that hydroxyl groups attached to ring A are essentials for BR activities. To verify this result, a series of analogs with a hydroxyl group at C-3 and similar substitutions in the benzoate group are currently under study. We hope that analyzing both sets of data will allow for us to establish a relationship between chemical structure and biological activity for these BR analogs.

It is worth emphasizing that the RLIT is the most widely used test and most of the proposed SARs are based on these data. However, it is also known that for the same BR, different activities can be obtained by using different bioassays. This occurs because BRs affect a variety of plant processes and consequently many plant responses can be obtained. Therefore, an SAR obtained by RLIT will probably not apply for BR activities in a BSI. For this reason, we used two different tests to evaluate the bioactivity in the synthesized analogs.

Finally, analog **41f** was the most active analog in both assays, even though these bioassays measure completely different effects on plant growth. Therefore, an exploratory docking study in the binding site of crystallized protein BRI1-BAK1 was carried out. The results show that the binding energy and orientation adopted by docked analog **41f** is like that found for brassinolide (Appendix A). Similar results have been observed in previous reports for other analogs with aromatic ring substitutions [13,14,19].

## 3. Materials and Methods

### 3.1. General Chemicals and Methods

All reagents were purchased from commercial suppliers and used without further purification. Melting points were measured on an SMP3 apparatus (Stuart-Scientific, now Merck KGaA, Darmstadt, Germany) and are uncorrected. ^1^H-, ^13^C-, ^13^C DEPT-135, gs 2D HSQC and gs 2D HMBC NMR spectra were recorded in CDCl_3_ solutions, and are referenced to the residual peaks of CHCl_3_ at δ = 7.26 ppm and δ = 77.00 ppm for ^1^H and ^13^C, respectively, on an Avance Neo 400 Digital NMR spectrometer (Bruker, Rheinstetten, Germany) operating at 400.1 MHz for ^1^H and 100.6 MHz for ^13^C. Chemical shifts are reported in δ ppm and coupling constants (*J*) are given in Hz; multiplicities are reported as follows: singlet (s), doublet (d), broad doublet (bd), doublet of doublets (dd), doublet of triplets (dt), triplet (t), broad triplet (bt), quartet (q), doublet of quartet (dq), doublet of double doublets (ddd), triplet of triplets (tt) and multiplet (m). IR spectra were recorded as KBr disks in a FT-IR 6700 spectrometer (Nicolet, Thermo Scientific, San Jose, CA, USA) and frequencies are reported in cm^−1^. High-resolution mass spectra (HRMS-ESI) were recorded in a Bruker Daltonik. The analysis for the reaction products was performed with the following relevant parameters: dry temperature, 180 °C; nebulizer 0.4 Bar; dry gas, 4 L/min; and spray voltage, 4.5 kV in positive mode. The accurate mass measurements were performed at a resolving power of 140,000 FWHM at a range *m*/*z* 50–1300. For analytical TLC, silica gel 60 in a 0.25 mm layer was used and TLC spots were detected by heating after spraying with 25% H_2_SO_4_ in H_2_O. Chromatographic separations were carried out by conventional column on silica gel 60 (230–400 mesh) using hexane–EtOAc mixtures of increasing polarity. All organic extracts were dried over anhydrous magnesium sulfate and evaporated under reduced pressure, below 40 °C.

### 3.2. Synthesis

#### 3.2.1. 3,6-Dioxo-5β-Cholan-24-Oic Acid (**42**)

A solution of hyodeoxycholic acid (**39**) (10.0 g, 25.47 mmol) in 800 mL of acetone was cooled in an ice-water bath at 10 °C with constant stirring. Then, 40 mL of Jones reagent solution (6.5 g CrO_3_, 6 mL H_2_SO_4_ and 18 mL H_2_O) was added. The reaction mixture was kept under constant stirring at 10 °C for 4 h. (TLC). Then, 80 [mL] of CH_3_OH was added to destroy the excess oxidizing agent. The resulting mixture was evaporated and diluted with water (400 mL), extracted with CH_2_Cl_2_ (300 mL) and washed with water (3 × 100 mL). The organic layer was dried over anhydrous MgSO_4_ and filtered, and the solvent was removed by distillation in a rotary evaporator. Compound **42** (9.7 g, 97% yield) was a colorless solid (m.p. = 164.4–164.7 °C). IR_νmax_ (KBr, cm^−1^): 3432–2650 (OH); 2964 and 287 (C-H); 1717 (C=O); 1466 (CH_2_-); 1387 (CH_3_-); 1245 and 1167 (C-O). ^1^H NMR (400.1 MHz, CDCl_3_): δ (ppm) = 2.64 (1H, dd, *J* = 14.3 and 13.7 Hz, H-4α); 2.48 (1H, dd, *J* = 12.9 and 4.7 Hz, H-5); 0.955 (3H, s, H-19); 0.945 (3H, d, *J* = 6.7 Hz, H-21); 0.697 (3H, s, H-18). ^13^C NMR (100.6 MHz, CDCl_3_): δ (ppm) = 210.92 (C-6); 208.76 (C-3); 179.44 (C-24); 59.69 (C-5); 56.76 (C-14); 55.77 (C-17); 43.08 (C-13); 42.13 (C-7); 40.89 (C-9); 39.87 (C-4); 39.47 (C-12); 38.28 (C-10); 36.67 (C-20); 36.47 (C-1); 35.74 (C-2); 35.19 (C-8); 30.84 (C-23); 30.60 (C-22); 27.93 (C-16); 23.89 (C-15); 22.45 (C-19); 21.28 (C-21); 11.98 (C-18) (Appendix A).

#### 3.2.2. 24-Nor-5β-Chol-22-Ene-3,6-Dione (**43**)

To a solution of **42** (4.00 g, 10.30 mmol) in dry benzene (120 mL), Cu(OAc)_2_*H_2_O (400 mg, 2.0 mmol) and pyridine (2.5 mL) were added. Then, under reflux, Pb(OAc)_4_ (11.2 g, 12.63 mmol) was added in four portions at hourly intervals. After the addition was completed, the reaction was continued for 4 h. The end of the reaction was verified by TLC; then, the mixture was filtered, and the solvent was evaporated under reduced pressure. The crude was re-dissolved in AcOEt (60 mL) washed with saturated aqueous NaCl solution (3 × 60 mL) and chromatographed on silica gel with hexane–EtOAc mixtures of increasing polarity (19.8:0.2→15.8:4.2). Compound **43** (1.71 g, 42.7% yield) was obtained as a colorless solid, m.p. = 180.5–182.4 °C (AcOEt/hex.), (177.5–178.9 °C [25] and 197–200 °C [29]). IR_νmax_ (KBr, cm^−1^): 3073 (CH=CH_2_); 2964, 2947, 2873 and 2855 (C-H); 1716 (C=O); 1693 (C=O); 1632 (C=C); 1466 (CH_2_-); 1382 (CH_3_-); 1245 and 1216 (C-O); 908 (CH=CH_2_). ^1^H NMR (400.1 MHz, CDCl_3_), δ (ppm) = 5.65 (1H, ddd, *J* = 17.2, 10.1 and 8.5 Hz, H-22); 4.91 (1H, dd, *J* = 17.2 and 1.9 Hz, H*_trans_*-23); 4.83 (1H, dd, *J* = 10.1 and 1.9 Hz, H*_cis_*-23); 2.64 (1H, dd, *J* = 14.7 and 13.2 Hz, H-4α); 2.47 (1H, dd, *J* = 12.5 and 4.9 Hz, H-5); 2.39 (1H, dd, *J* = 14.0 and 5.0 Hz, H-1α); 1.11 (1H, ddd, *J* = 12.1, 11.9 and 6.0 Hz, H-15); 1.04 (3H, d, *J* = 6.7 Hz, H-21); 0.954 (3H, s, H-19); 0.715 (3H, s, H-18). ^13^C NMR (100.6 MHz, CDCl_3_), δ (ppm) = 210.81 (C-6); 208.63 (C-3); 144.69 (C-22); 111.96 (C-23); 59.71 (C-5); 56.81 (C-14); 55.36 (C-17); 42.99 (C-13); 42.13 (C-7); 41.06 (C-20); 40.93 (C-9); 39.87 (C-4); 39.37 (C-12); 38.27 (C-10); 36.66 (C-8); 36.46 (C-1); 35.74 (C-2); 28.18 (C-16); 23.90 (C-15); 22.45 (C-19); 21.27 (C-11); 20.06 (C-21); 12.13 (C-18). IR, ^1^H and ^13^C NMR spectroscopic data were consistent with those reported [24] (Appendix A).

#### 3.2.3. 24-Nor-5α-Chol-22-Ene-3,6-Dione (**44**)

Compound **43** (2.00 g, 5.84 mmol) was dissolved in 195 mL of 2.5% *v*/*v* HCl-MeOH at reflux and constant agitation for 2 h. The end of the reaction was verified by TLC. The solvent was evaporated under reduced pressure, and the crude was re-dissolved in 60 mL of EtOAc. The organic layer was washed with a saturated solution of NaHCO_3_ (3 × 60 mL) and water (2 × 30 mL), dried over MgSO_4_ and filtered. The solvent was evaporated under reduced pressure. The crude was re-dissolved in CH_2_Cl_2_ (5 mL) and chromatographed on silica gel with hexane–EtOAc mixtures of increasing polarity (9.8:0.2→4.0:6.0). Compound **44** (1.88 g, 94.0% yield) was obtained as a colorless solid, m.p. = 148.9–149.7 °C (AcOEt/hex.). IR_νmax_ (KBr, cm^−1^): 3073 (CH=CH_2_); 2946, 2870 and 2866 (C-H); 1712 (C=O); 1638 (C=C); 1460 (CH_2_-); 1389 (CH_3_-); 1248 and 1216 (C-O); 906 (CH=CH_2_). ^1^H NMR (400.1 MHz, CDCl_3_), δ (ppm) = 5.65 (1H, ddd, *J* = 17.2, 10.2 and 8.5 Hz, H-22); 4.91 (1H, dd, *J* = 17.2 and 1.8 Hz, H*_trans_*-23); 4.83 (1H, dd, *J* = 10.2 and 1.8 Hz, H*_cis_*-23); 2.62–2.54 (2H, m, H-5 and H-2); 2.00 (1H, t, *J* = 13.0 Hz, H-7β); 1.43 (1H, ddd, *J* = 16.7, 12.7 and 3.7 Hz, H-11β); 1.10 (1H, ddd, *J* = 18.0, 11.7 and 5.9 Hz, H-15); 1.03 (d, *J* = 6.6 Hz, 3H, H-21); 0.953 (s, 3H, H-19); 0.712 (s, 3H, H-18). ^13^C NMR (100.6 MHz, CDCl_3_), δ (ppm) = 211.23 (C-3); 209.03 (C-6); 144.75 (C-22); 111.92 (C-23); 57.47 (C-5); 56.58 (C-14); 55.34 (C-9); 53.44 (C-17); 46.55 (C-7); 42.95 (C-13); 41.22 (C-10); 41.06 (C-20); 39.22 (C-12); 38.06 (C-1); 37.96 (C-8); 37.35 (C-2); 36.96 (C-4); 28.15 (C-16); 23.95 (C-15); 21.62 (C-11); 20.03 (C-21); 12.54 (C-19); 12.17 (C-18) (Appendix A).

#### 3.2.4. 22(S), 23-Dihydroxy-24-Nor-5α-Cholan-3,6-Dione (**40**) and 22(R), 23-Dihydroxy-24-Nor-5α-Cholan-3,6-Dione (**40a**)

To a solution of alkene **44** (2.50 g, 7.30 mmol) in acetone (150 mL), *N*-Methylmorpholine *N*-oxide (NMO) (0.45 g, 3.84 mmol) was added. Then, the mixture was homogenized by magnetic stirring and 2.0 mL of 4% OsO_4_ (0.210 mmol) was added dropwise with stirring for 36 h at room temperature. The end of the reaction was verified by TLC. Then, the solvent was removed (until a 25 mL approximate volume), and water (25 mL) and Na_2_S_2_O_3_.5H_2_O (25 mL saturated solution) were added. The organic layer was extracted with EtOAc (2 × 30 mL), washed with saturated NaHCO_3_ (3 × 60 mL) and water (2 × 20 mL), dried over MgSO_4_ and filtered. The solvent was evaporated under reduced pressure. The crude was re-dissolved in CH_2_Cl_2_ (10 mL) and chromatographed on silica gel with hexane–EtOAc mixtures of increasing polarity (19.8:0.2→9.8:10.2). A mixture of **40**/**40a** = 4.0/1.0 was obtained as a colorless solid (1.61 g, 64.3% yield).

#### 3.2.5. 22(S)-Hydroxy-24-Nor-5α-Cholan-3,6-Dioxo-(4-Substituted)-Benzoate-23-yl (**41a**–**41f**)

**General procedure**: A mixture of **40**/**40a** was dissolved in pyridine (py, 15 mL). Later, 4-dimetilaminopiridina (DMAP) (40 mg) and *p*-PhCOCl were added with slow stirring at 5–10 °C. The end of the reaction was verified by TLC (3 h). Then, 3 mL of water (60 °C) was added and stirred for 20 min. The mixture was extracted with EtOAc (2 × 25 mL), and the organic layer was washed with 5% KHSO_4_ (2 × 5 mL) and water (2 × 10 mL), dried over MgSO_4_ and filtered. The solvent was evaporated under reduced pressure. The crude was redissolved in CH_2_Cl_2_ (5 mL) and chromatographed on silica gel with hexane–EtOAc mixtures of increasing polarity.

#### 3.2.6. 22(S)-Hydroxy-24-Nor-5α-Cholan-3,6-Dioxobenzoate-23-yl (**41a**)

Mixture **40**/**40a** (324.0 mg, 0.86 mmol), py (15 mL), DMAP (40 mg) and PhCOCl (392 μL, 2.79 mmol, d = 1.21 g/mL). Compound **41a** (107 mg, 25.9% yield) was obtained as a colorless solid, m.p. = 206.4–207.8 °C (EtOAc/hex). IR_νmax_ (KBr, cm^−1^): 3446 (O-H); 2947 (CH_3_-); 2871 (CH_2_-); 1705 (C=O); 1454 (CH_2_-); 1274 and 1024 (C-O); 718 (Ar-H). ^1^H NMR (400.1 MHz, CDCl_3_): δ (ppm) = 8.05 (2H, dd, *J* = 7.8 and 1.4 Hz, 2H, H-2’ and H-6’); 7.58 (1H, t, *J* = 7.4 Hz, H-4’); 7.46 (2H, t, *J* = 7.8 Hz, H-3’ and H-5’); 4.49 (1H, dd, *J* = 11.5 and 1.9 Hz, H-23a); 4.20 (1H, dd, *J* = 11.5 and 9.1 Hz, H-23b); 4.06 (1H, ddd, *J* = 9.1, 3.6 and 1.9 Hz, H-22); 2.63–2.54 (2H, m, H-5 and H-2); 2.01 (1H, t, *J* = 12.6 Hz, H-7β); 1.18 (1H, ddd, *J* = 12.5, 12.0 and 6.0 Hz, H-15); 1.06 (d, *J* = 6.8 Hz, 3H, H-21); 0.959 (s, 3H, H-19); 0.732 (s, 3H, H-18). ^13^C NMR (100.6 MHz, CDCl_3_): δ (ppm) = 211.20 (C-3); 208.92 (C-6); 167.00 (Ar-CO_2_); 133.25 (C-4’); 129.80 (C-1’); 129.62 (C-2’ and C-6’); 128.45 (C-3’ and C-5’); 71.74 (C-22); 66.35 (C-23); 57.44 (C-5); 56.19 (C-14); 53.34 (C-9); 52.83 (C-17); 46.50 (C-7); 43.42 (C-13); 41.17 (C-10); 40.27 (C-20); 39.27 (C-12); 38.01 (C-1); 37.95 (C-8); 37.33 (C-2); 36.94 (C-4); 27.39 (C-16); 24.05 (C-15); 21.61 (C-11); 12.89 (C-21); 12.53 (C-19); 11.80 (C-18) (Appendix A). HRMS-ESI (positive mode): *m*/*z* calculated for C_30_H_40_O_5_: 481.2949 [M + H]^+^; found 481.2918 [M + H]^+^ (Appendix A).

#### 3.2.7. 22(S)-Hydroxy-24-Nor-5α-Cholan-3,6-Dioxo-(4-Methyl)-Benzoate-23-yl (**41b**)

Mixture **40**/**40a** (200 mg, 0.53 mmol), py (10 mL), DMAP (70 mg) and *p*-CH_3_-PhCOCl (287 μL, 0.68 mmol, d = 1.17 g/mL). Compound **41b** (22.1 mg, 8.4% yield) was obtained as a colorless solid, m.p. = 217.9–219.3 °C (EtOAc/hex). IR_νmax_ (KBr, cm^−1^): 3446 (O-H); 2950 (CH_3_-); 2905 (CH_2_-); 2870 (CH-); 1720 (C=O); 1705 (C=O); 1609 (C=C); 1391 (CH_3_-); 1277 and 1095 (C-O); 757 (Ar-H). ^1^H NMR (400.1 MHz, CDCl_3_): δ (ppm) = 7.95 (2H, d, *J* = 8.1 Hz, H-2’ and H-6’); 7.27 (2H, d, *J* = 8.1 Hz, H-3’ and H-5’); 4.49 (1H, dd, *J* = 11.5 and 1.7 Hz, H-23a); 4.20 (1H, dd, *J* = 11.5 and 9.1 Hz, H-23b); 4.08–4.05 (1H, m, H-22); 2.65–2.56 (2H, m, H-5 and H-2); 2.44 (3H, s, CH_3_-Ar); 2.03 (1H, t, *J* = 12.8 Hz, H-7β); 1.19 (1H, ddd, *J* = 12.8, 11.7 and 5.7 Hz, H-15); 1.08 (3H, d, *J* = 6.8 Hz, H-21); 0.978 (3H, s, H-19); 0.750 (3H, s, H-18). ^13^C NMR (100.6 MHz, CDCl_3_): δ (ppm) = 211.18 (C-3); 208.90 (C-6); 166.06 (Ar-CO_2_); 143.97 (C-4’); 129.64 (C-3’ and C-5’); 129.14 (C-2’ and C-6’); 127.04 (C-1’); 71.72 (C-22); 66.17 (C-23); 57.42 (C-5); 56.18 (C-14); 53.34 (C-9); 52.81 (C-17); 46.48 (C-7); 43.40 (C-13); 41.15 (C-10); 40.23 (C-20); 39.25 (C-12); 37.99 (C-1); 37.94 (C-8); 37.31 (C-2); 36.92 (C-4); 27.37 (C-16); 24.04 (C-15); 21.65 (CH_3_-Ar); 21.60 (C-11); 12.88 (C-21); 12.52 (C-19); 11.79 (C-18) (Appendix A). HRMS-ESI (positive mode): *m*/*z* calculated for C_31_H_42_O_5_: 495.3105 [M + H]^+^; found 495.3119 [M + H]^+^ (Appendix A).

#### 3.2.8. 22(S)-Hydroxy-24-Nor-5α-Cholan-3,6-Dioxo-(4-Methoxy)-Benzoate-23-yl (**41c**)

Mixture **40**/**40a** (211.3 mg, 0.56 mmol), py (10 mL), DMAP (45 mg) and *p*-CH_3_O-PhCOCl (373 mg, 0.73 mmol). Compound **41c** (87.7 mg, 30.6% yield) was obtained as a colorless solid, m.p. = 202.3–204.2 °C (EtOAc/hex). IR_νmax_ (KBr, cm^−1^): 3490 (O-H); 2967 (CH_3_-); 2890 (CH_2_-); 1707 (C=O); 1607 (C=C); 1514 (C=C); 1425 (CH_2_-); 1257 and 1075 (C-O); 771 (Ar-H). ^1^H NMR (400.1 MHz, CDCl_3_): δ (ppm) = 7.99 (2H, d, *J* = 8.9 Hz, H-2’ and H-6’); 6.91 (2H, d, *J* = 8.9 Hz, H-3’ and H-5’); 4.45 (1H, dd, *J* = 11.5 and 1.9 Hz, H-23a); 4.15 (1H, dd, *J* = 11.5 and 9.1 Hz, H-23b); 4.04–4.00 (1H, m, H-22); 3.85 (3H, s, OCH_3_); 2.62–2.53 (2H, m, H-5 and H-2); 2.00 (1H, t, *J* = 13.1 Hz, H-7β); 1.16 (1H, ddd, *J* = 12.3, 11.5 and 5.7 Hz, H-15); 1.04 (3H, d, *J* = 6.9 Hz, H-21); 0.943 (3H, s, H-19); 0.715 (3H, s, H-18). ^13^C NMR (100.6 MHz, CDCl_3_): δ (ppm) = 211.19 (C-3); 208.91 (C-6); 166.73 (Ar-CO_2_); 163.53 (C-4’); 131.66 (C-2’ and C-6’); 122.14 (C-1’); 113.65 (C-3’ and C-5’); 71.74 (C-22); 66.05 (C-23); 57.39 (C-5); 56.15 (C-14); 55.42 (C-9); 53.31 (C-17); 52.79 (C-17); 46.46 (C-7); 43.38 (C-13); 41.14 (C-10); 40.23 (C-20); 39.23 (C-12); 37.97 (C-1); 37.92 (C-8); 37.29 (C-2); 36.90 (C-4); 27.36 (C-16); 24.03 (C-15); 21.58 (C-11); 12.87 (C-21); 12.50 (C-19); 11.77 (C-18) (Appendix A). HRMS-ESI (positive mode): *m*/*z* calculated for C_31_H_42_O_6_: 511.3054 [M + H]^+^; found 511.3071 [M + H]^+^ (Appendix A).

#### 3.2.9. 22(S)-Hydroxy-24-Nor-5α-Cholan-3,6-Dioxo-(4-Fluoro)-Benzoate-23-yl (**41d**)

Mixture **40**/**40a** (160.6 mg, 0.43 mmol), py (10 mL), DMAP (50 mg) and *p*-F-PhCOCl (160 μL, 1.35 mmol, d = 1.34 g/mL). Compound **41d** (49.6 mg, 23.3% yield) was obtained as a colorless solid, m.p. = 189.3–190.1 °C (EtOAc/hex). IR_νmax_ (KBr, cm^−1^): 3505 (O-H); 2967 (CH_3_-); 2867 (CH_2_-); 1716 (C=O); 1600 (C=C); 1508 (C=C); 1455 (CH_2_-); 1380 (CH_3_-); 1287 and 1071 (C-O); 768 (Ar-H). ^1^H NMR (400.1 MHz, CDCl_3_): δ (ppm) = 8.06 (2H, dd, *J* = 8.7 and 5.4 Hz, H-2’ and H-6’); 7.12 (2H, t, *J* = 8.7 Hz, H-3’ and H-5’); 4.47 (1H, dd, *J* = 11.4 and 1.9 Hz, H-23a); 4.18 (1H, dd, *J* = 11.3 and 9.1 Hz, H-23b); 4.05–4.02 (1H, m, H-22); 2.62–2.53 (2H, m, H-5 and H-2); 2.00 (1H, t, *J* = 12.7 Hz, H-7β); 1.17 (1H, ddd, *J* = 12.1, 11.5 and 5.8 Hz, H-15); 1.05 (3H, d, *J* = 7.0 Hz, H-21); 0.951 (3H, s, H-19); 0.723 (3H, s, H-18). ^13^C NMR (100.6 MHz, CDCl_3_): δ (ppm) = 211.18 (C-3); 208.88 (C-6); 166.01 (Ar-CO_2_); 165.89 (d, ^1^*J*_CF_ = 255.0 Hz, C-4’); 132.20 (d, ^3^*J*_CF_ = 9.3 Hz, C-2’ and C-6’); 126.05 (d, ^4^*J*_CF_ = 2.7 Hz, C-1’); 115.61 (d, ^2^*J*_CF_ = 22.0 Hz, C-3’ and C-5’); 71.69 (C-22); 66.45 (C-23); 57.2 (C-5); 56.17 (C-14); 53.33 (C-9); 52.82 (C-17); 46.47 (C-7); 43.41 (C-13); 41.15 (C-10); 40.35 (C-20); 39.25 (C-12); 37.99 (C-1); 37.93 (C-8); 37.31 (C-2); 36.91 (C-4); 27.37 (C-16); 24.04 (C-15); 21.60 (C-11); 12.87 (C-21); 12.52 (C-19); 11.79 (C-18) (Appendix A). HRMS-ESI (positive mode): *m*/*z* calculated for C_30_H_39_FO_5_: 499.2854 [M + H]^+^; found 499.2839 [M + H]^+^ (Appendix A).

#### 3.2.10. 22(S)-Hydroxy-24-Nor-5α-Cholan-3,6-Dioxo-(4-Chloro)-Benzoate-23-yl (**41e**)

Mixture **40**/**40a** (120.0 mg, 0.32 mmol), py (10 mL), DMAP (35 mg) and *p*-Cl-PhCOCl (485 μL, 1.28 mmol, d = 1.36 g/mL). Compound **41e** (32.7 mg, 19.9% yield) was obtained as a colorless solid, m.p. = 196.9–198.8 °C (EtOAc/hex). IR_νmax_ (KBr, cm^−1^): 3466 (O-H); 2970 (CH_3_-); 2869 (CH_2_-); 1709 (C=O); 1596 (C=C); 1487 (CH_2_-); 1281 and 1016 (C-O); 763 (Ar-H). ^1^H NMR (400.1 MHz, CDCl_3_): δ (ppm) = 8.00 (2H, d, *J* = 8.8 Hz, H-2’ and H-6’); 7.45 (2H, d, *J* = 8.8 Hz, H-3’ and H-5’); 4.50 (1H, dd, *J* = 11.6 and 2.1 Hz, H-23a); 4.20 (1H, dd, *J* = 11.6 and 9.2 Hz, H-23b); 4.07 (1H, ddd, *J* = 9.1, 3.8 and 2.1 Hz, H-22); 2.68–2.57 (2H, m, H-5 and H-2); 2.04 (1H, t, *J* = 12.8 Hz, H-7β); 1.20 (1H, ddd, *J* = 12.5, 11.7 and 5.8 Hz, H-15); 1.08 (3H, d, *J* = 7.0 Hz, H-21); 0.985 (3H, s, H-19); 0.756 (3H, s, H-18). ^13^C NMR (100.6 MHz, CDCl_3_): δ (ppm) = 211.19 (C-3); 208.88 (C-6); 166.12 (Ar-CO_2_); 139.70 (C-4’); 131.01 (C-2’ and C-6’); 128.25 (C-3’ and C-5’); 128.25 (C-1’); 71.66 (C-22); 66.55 (C-23); 57.43 (C-5); 56.18 (C-14); 53.34 (C-9); 52.83 (C-17); 46.48 (C-7); 43.41 (C-13); 41.16 (C-10); 40.38 (C-20); 39.26 (C-12); 38.00 (C-1); 37.93 (C-8); 37.31 (C-2); 36.92 (C-4); 27.38 (C-16); 24.04 (C-15); 21.60 (C-11); 12.87 (C-21); 12.52 (C-19); 11.80 (C-18) (Appendix A). HRMS-ESI (positive mode): *m*/*z* calculated for C_30_H_39_ClO_5_: 515.2559 [M + H]^+^; found 515.2538 [M + H]^+^ (Appendix A).

#### 3.2.11. 22(S)-Hydroxy-24-Nor-5α-Cholan-3,6-Dioxo-(4-Bromo)-Benzoate-23-yl (**41f**)

Mixture **40**/**40a** (206.0 mg, 0.55 mmol), py (10 mL), DMAP (62.8 mg) and *p*-Br-PhCOCl (921 mg, 1.65 mmol). Compound **41f** (45.5 mg, 14.9% yield) was obtained as a colorless solid, m.p. = 206.1–208.0 °C (EtOAc/hex). IR_νmax_ (KBr, cm^−1^): 3482 (O-H); 2957 (CH_3_-); 2890 (CH_2_-); 1722 (C=O); 1690 (C=O); 1591 (C=C); 1462 (CH_2_-); 1398 (CH_3_-); 1272 and 1071 (C-O); 760 (Ar-H). ^1^H NMR (400.1 MHz, CDCl_3_): δ (ppm) = 7.91 (2H, d, *J* = 8.5 Hz, H-2’ and H-6’); 7.59 (2H, d, *J* = 8.5 Hz, H-3’ and H-5’); 4.48 (1H, dd, *J* = 11.7 and 1.9 Hz, H-23a); 4.19 (1H, dd, *J* = 11.7 and 9.1 Hz, H-23b); 4.04 (1H, ddd, *J* = 8.9, 3.4 and 2.5 Hz, H-22); 2.63–2.55 (2H, m, H-5 and H-2); 2.01 (1H, t, *J* = 12.5 Hz, H-7β); 1.18 (1H, ddd, *J* = 12.1, 11.6 and 5.8 Hz, H-15); 1.05 (3H, d, *J* = 6.9 Hz, H-21); 0.959 (3H, s, H-19); 0.730 (s, 3H, H-18). ^13^C NMR (100.6 MHz, CDCl_3_): δ (ppm) = 211.19 (C-3); 208.88 (C-6); 166.28 (Ar-CO_2_); 131.81 (C-2’ and C-6’); 131.14 (C-3’ and C-5’); 128.71 (C-1’); 128.41 (C-4’); 71.68 (C-22); 66.58 (C-23); 57.44 (C-5); 56.19 (C-14); 53.36 (C-9); 52.84 (C-17); 46.49 (C-7); 43.43 (C-13); 41.17 (C-10); 40.39 (C-20); 39.27 (C-12); 38.02 (C-1); 37.94 (C-8); 37.32 (C-2); 36.93 (C-4); 27.39 (C-16); 24.04 (C-15); 21.61 (C-11); 12.88 (C-21); 12.54 (C-19); 11.81 (C-18) (Appendix A). HRMS-ESI (positive mode): *m*/*z* calculated for C_30_H_39_BrO_5_: 561.2038 [M + H]^+^; found 561.1994 [M + H]^+^ (Appendix A).

#### 3.2.12. 22(S), 23-Dihydroxy-24-Nor-5α-Cholan-3,6-Dione (**40**) from **41a**

To a solution of benzoate **41a** (60 mg, 0.13 mmol) in methanol (20 mL) and water (10 mL), K_2_CO_3_ (63.2 mg, 0.46 mmol) was added. Then the mixture was homogenized by magnetic stirring and reflux for 3 h (TLC). The reaction mixture was cooled in an ice-water bath (0–5 °C) then, 50 mL HCl (5% *w*/*w*) were added until the formation of a white precipitate. The organic layer was extracted with EtOAc (2 × 60 mL), washed with water (2 × 50 mL), dried over MgSO_4_, and filtered. The solvent was evaporated under reduced pressure. The crude was re-dissolved in CH_2_Cl_2_ (10 mL) and chromatographed on silica gel with hexane–EtOAc (3:7 ratio). Compound **40** (17 mg, 36.2% yield) was obtained as a colorless solid, m.p. = 226.3–227.2 °C (ether/MeOH). IR_νmax_ (KBr, cm^−1^): 3510 (O-H); 2968 (CH_3_-); 2935 (CH_2_-); 2865 (CH-); 1705 (C=O); 1428 (CH_2_-); 1381 (CH_3_-); 1264 and 1064 (C-O). ^1^H NMR (400.1 MHz, CDCl_3_): δ (ppm) = 3.81 (1H, dt, *J* = 9.6 and 3.1 Hz, H-22); 3.65 (1H, dd, *J* = 11.3 and 2.3 Hz, H-23a); 3.54 (1H, dd, *J* = 11.3 and 9.6 H, H-23b); 2.63–2.54 (2H, m, H-5 and H-2); 2.00 (1H, t, *J* = 12.4 Hz, H-7β); 0.965 (3H, d, *J* = 6.8 Hz, H-21); 0.956 (3H, s, H-19); 0.706 (3H, s, H-18). ^13^C NMR (100.6 MHz, CDCl_3_): δ (ppm) = 211.24 (C-3); 208.92 (C-6); 73.82 (C-22); 66.48 (C-23); 57.55 (C-5); 56.20 (C-14); 53.43 (C-9); 52.81 (C-17); 46.58 (C-7); 43.44 (C-13); 41.22 (C-10); 40.01 (C-20); 39.31 (C-12); 38.10 (C-1); 38.00 (C-8); 37.37 (C-2); 36.99 (C-4); 27.39 (C-16); 24.11 (C-15); 21.67 (C-11); 13.09 (C-21); 12.58 (C-19); 11.78 (C-18) (Appendix A). HRMS-ESI (positive mode): *m*/*z* calculated for C_23_H_36_O_4_: 377.2686 [M + H]^+^; found 377.2672 [M + H]^+^ (Appendix A).

### 3.3. Biological Activity

#### 3.3.1. Bioactivity in the Rice Lamina Inclination Test (RLIT)

The bioactivity of BR analogs (**40**, and **41a**–**41f**) was evaluated by RLIT [31] and using a modified procedure previously described [19]. Seeds of a local rice cultivar (*Oryza sativa* L.) of the *Zafiro* variety (provided by INIA-QUILAMAPU, Chillán, Ñuble, Chile) were sterilized, sown and grown for about 10 days in pots with substrate, maintained at 22 °C, with a 16 h light/8 h dark photoperiod, and 50–60% relative humidity in a plant growth chamber. Once the rice plants presented the second internode of the lamina, an 8 cm segment was cut. These segments were then placed in a Petri dish containing sterile distilled water (60 mL) with the amounts of BR analogs required to reach the treatment concentration (1 × 10^−8^; 1 × 10^−7^ and 1 × 10^−6^ M). Brassinolide (APExBIO) at the same tested concentrations was used as positive control, whereas the negative control was an aqueous solution with the same volume of dimethyl sulfoxide (DMSO, 1%) used to dilute the BR analogs. All treatments were incubated for 72 h at 25 °C in the dark to finally measure the angle of inclination in the unrolled sheet between the leaf and the sheath. Each treatment consisted of 8 independent replicates. The effect produced in the negative control was subtracted from each treatment. Mean values with at least significant difference (*p* < 0.05; Student’s *t*-test) were considered using Excel software (Microsoft Corporation, (Microsoft Corporation, WA, USA)). Images were taken with a Leica EZ4HD stereo microscope with camera software LAS EZ (Leica Microsystems, Wetzlar, Alemania).

#### 3.3.2. Bean Second Internode Bioassay

The bean second internode test for compounds **40** and **41a**–**41f** was carried out using a previously reported procedure [32], with some modifications [18]. Bean seeds (*Phaseolus vulgaris* L., cv. Pinto) were sown and germinated for three days, and subsequently were transplanted into pots containing perlite, vermiculite and substrate. The pots were kept in a plant growth chamber at 22 °C, with 48 W/m^2^ light at a 16 h/8 h light/dark photoperiod. When the second internode reached 1–2 mm long, the bean plants were treated with tested compounds dissolved in DMSO (1%) and water at 1 × 10^−8^ M concentration, by adding them into a small scar generated once the bract was removed from the base of the second internode. At the time of application, a 5 µL drop of each solution was mixed with a 2 µL drop of TWEEN^®^ 20 (AMRESCO^®^) for adhesion. Control plants were treated with water and TWEEN^®^ 20 only. Measurements of second internode length were made after 5 days. The difference between the length of the second internode of treated and control plants was used as a measure of activity. Mean values with at least significant difference (*p* < 0.05; Student’s *t*-test) were considered using Excel software 2021 (Microsoft Corporation, WA, USA).

#### 3.3.3. Molecular Docking Study

The docking procedure involved positioning brassinolide (**1**) and synthetic compound **41f** within the active site of the crystallized protein BRI1-BAK1 (PDB: 4m7e) using the AutoDock Vina program. The protocol used to carry out this study was reported previously [19].

## 4. Conclusions

Herein, a short synthesis route (five steps) was developed, which, starting from hyodeoxycholic acid (**39**), allowed for obtaining compound **40** and five new brassinosteroid analogs with 3,6 dioxo function, 24-Nor-22(*S*)-hydroxy side chain and *p*-substituted benzoate function at C-23 (compounds **41a**–**f**). This effort is part of a more general task in which the final goal is to establish a relationship between the chemical structure of BR analogs and their biological activity. With this aim, synthesized BR analogs were tested for activity using two different assays, namely, the rice lamina inclination test and the bean second-node bioassay, and the results indicate that compounds **40** and **41f** are the more active analogs at 1 × 10^−8^ M concentration. RLIT activities were correlated with inductive and size effects of substituents at the *para* position in the benzoate group. A comparison of these results with those previously obtained for **37** and **38** seems to indicate that the effect of the electronic density and size of the alkyl chain on biological activities depends on the entire BR structure, and that hydroxyl groups attached to ring A are essentials for BR activities. More data are needed to verify this conclusion before it can be used as a structural requirement for bioactivity in brassinosteroid analogs.

On the other hand, much more work is needed to establish a relationship between data obtained by using different bioassays. It is also necessary to find a way to define which bioassay gives the best indication of BR activities in each application.

Finally, it has become common to explain BR activities by molecular docking studies in which new molecules are docked to the binding site of natural brassinolide on *Arabidopsis Thaliana*. In light of these results, it is unclear how docking studies can explain BR activities occurring by different mechanisms.

## Figures and Tables

**Figure 1 ijms-25-07515-f001:**
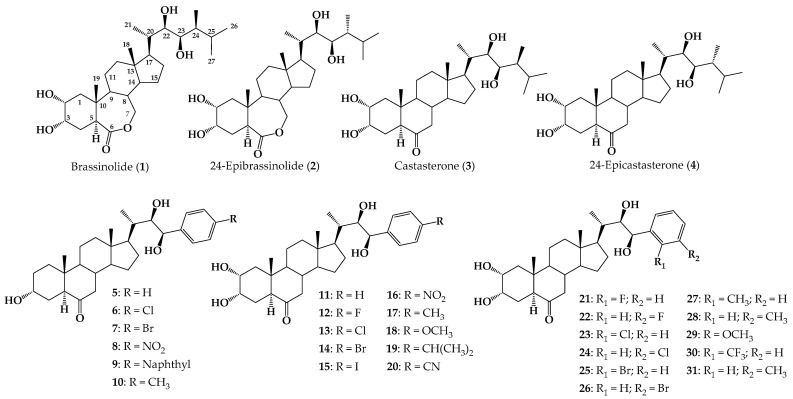
Structure of natural occurring BRs **1**–**4**, and synthetic arylbrassinosteroid analogs **5**–**31**.

**Figure 2 ijms-25-07515-f002:**
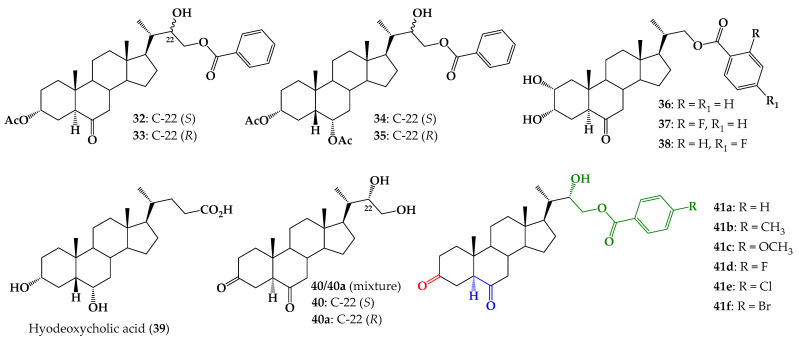
Structure of 24-norcholane type BR analogs with benzoate function at C-23 (compounds **32**–**35**), 23,24-bisnorcholanic analogs (compounds **36**–**38**), hyodeoxycholic acid, precursor mixture **40**/**40a** and new analogs 3,6-dioxo with benzoate function at C-23 (compounds **41a**–**f**).

**Figure 3 ijms-25-07515-f003:**
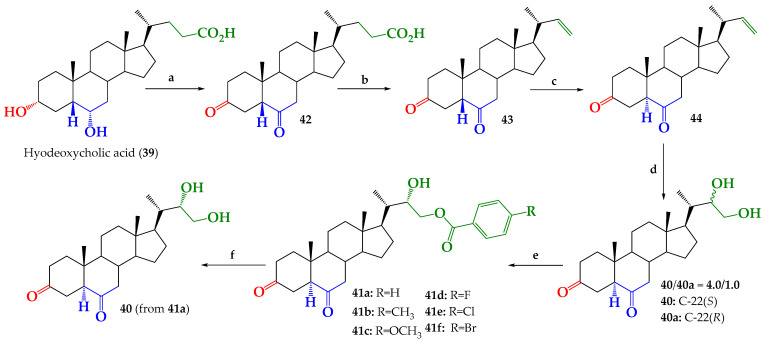
Synthesis of compounds **40**, **42**–**44**, epimeric mixture **40**/**40a** and new BR analogs **41a**–**f**. Conditions: **a**. Jones ((CrO_3_/H_2_SO_4_/(CH_3_)_2_CO), r.t. 4 h, 97.0%. **b**. Pb(OAc)_4_/Cu(OAc)_2_/C_5_H_5_N/C_6_H_6_, reflux, 4 h, 42.7%. **c**. HCl 2.5%, CH_3_OH, reflux, 2 h, 94.0%. **d**. Upjohn dihydroxylation (OsO_4_/NMMO/(CH_3_)_2_CO), r.t, 36 h, 64.3%. **e**. 4-R-PhCOCl/CH_2_Cl_2_/C_5_H_5_N/DMAP, 0–5 °C, 6 h, C.C separation and subsequent crystallization **41a**: 25.9%; **41b**: 8.4%; **41c**: 30.6%; **41d**: 23.3%; **41e**: 19.9%; **41f**: 14.9%. **f**. K_2_CO_3_/MeOH/reflux, 3 h, 36.2%.

**Table 1 ijms-25-07515-t001:** Main spectroscopic differences observed in ^1^H and ^13^C NMR spectra of compounds **43** and **44** indicating isomerization from 5β to 5α.

Compounds	^1^H NMR	^13^C NMR
**43**	δ = 2.47 ppm (1H, dd, *J* = 12.5 and 4.9 Hz, H-5)	δ = 22.45 (C-19)
**44**	δ = 2.62–2.54 (2H, m, H-5 and H-2)	δ = 12.54 (C-19)

**Table 2 ijms-25-07515-t002:** Effect of different concentrations of brassinolide (**1**) and 24-norcholane BR type analogs on lamina inclination of rice seedlings: 3,6-dioxo-23-dihydroxy (**40**); 2,6-dioxo-22(*S*)-hydroxy-23-(4-substituted)-benzoate (**41a**–**f**).

Bending Angles between Laminae and Sheaths (Degrees ± Standard Error) ^1^			
Compounds	1 × 10^−8^ M	1 × 10^−7^ M	1 × 10^−6^ M
**1**	81 ± 7.1 ^a^	88 ± 4.8 ^a^	90 ± 7.5 ^a^
**40**	26 ± 4.1 ^d^	38 ± 4.9 ^c^	13 ± 4.1 ^d^
**41a**	19 ± 4.8 ^e^	-	-
**41b**	19 ± 3.2 ^e^	32 ± 4.2 ^de^	32 ± 4.2 ^c^
**41c**	-	28 ± 2.6 ^e^	54 ± 8.4 ^b^
**41d**	38 ± 2.6 ^c^	34 ± 2.7 ^d^	-
**41e**	32 ± 4.2 ^d^	31 ± 8.2 ^de^	-
**41f**	75 ± 2.0 ^b^	49 ± 4.5 ^b^	33 ± 2.6 ^c^
**Negative Control**		16 ± 2.0	

^1^ Values represent the mean ± standard deviation of two independent experiments with at least eight replicates each. Average angle of negative control: 16 ± 2.0. Superscript letters represent experiments with a significant difference between treatments at 0.05 significance level (Student’s *t*-test). (−): values lower than negative control. Brassinolide (**1**) was used as positive control.

**Table 3 ijms-25-07515-t003:** Activity in bean second internode bioassay of brassinolide, **40**, and BR analogs **41a**–**f**, at 1 × 10^−8^ M.

Compounds	Prolongation of the Second Internode, mm ± SD, at Concentration 1 × 10^−8^ M
**1**	20.2 ± 0.6 ^a^
**40**	11.2 ± 7.8 ^b^
**41a**	4.4 ± 1.9 ^d^
**41b**	2.9 ± 1.9 ^e^
**41c**	9.8 ± 1.8 ^b^
**41d**	5.5 ± 1.8 ^d^
**41e**	8.5 ± 2.8 ^bc^
**41f**	10.8 ± 2.7 ^b^
**Negative Control**	1.0 ± 0.1 ^f^

Superscript letters indicate the level of significance according to student’s *t*-test *p* < 0.05. Brassinolide (**1**) was used as positive control.

## Data Availability

All data is available at Appendix A.

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
