# Peer review of "Novel Brassinosteroid Analogues with 3,6 Dioxo Function, 24-Nor-22(S)-Hydroxy Side Chain and p-Substituted Benzoate Function at C-23—Synthesis and Evaluation of Plant Growth Effects"

_ijms, 2024, doi:10.3390/ijms25147515_

Round 1

Reviewer 1 Report

Comments and Suggestions for Authors

In this paper, Jorquera et al. synthesized new brassinosteroid analogs obtained from hyodeoxycholic acid, with 3,6 dioxo function, 24-Nor-22(S)-hydroxy side chain and p-substituted benzoate function at C-23. The authors validated this compound in two separate bioassays.

This is a novel research which may benefit many fields, however, the validation of activity of the new compound is insufficient. Additional bioassays or similar bioassays in additional species is required.  

Author Response

Dear Reviewer 1

In this paper, Jorquera et al. synthesized new brassinosteroid analogs obtained from hyodeoxycholic acid, with 3,6 dioxo function, 24-Nor-22(S)-hydroxy side chain and p-substituted benzoate function at C-23. The authors validated this compound in two separate bioassays.

This is a novel research which may benefit many fields, however, the validation of activity of the new compound is insufficient. Additional bioassays or similar bioassays in additional species is required.

The reason why only two different bioassays were used is presented in a more detailed way at the end of the Discussion Section.

Reviewer 2 Report

Comments and Suggestions for Authors

The manuscript (Novel brassinosteroid analogs with 3,6 dioxo function, 24-Nor-22(S)-hydroxy side chain and p-substituted benzoate function at C-23. Synthesis and evaluation of plant growth effects)  deals with the synthesis of new brassinosteroid derivatives and their biological activity. Unfortunately, some important information is missing in the article and should be added. Therefore, I cannot recommend the paper for publication without a major revision.

1) The Results and Discussion section is very poorly written, the discussion is missing and therefore this section should be completely rewritten. (the results of the bioactivity tests are not discussed at all and their in-depth interpretation is also missing).

- Tables 2.2 and 2.3 should also include results for negative controls.

- What was the final concentration of DMSO in the bioassays?

- Why was only 1 concentration (10-8M) tested for the biotest on the second bean internodes, in my opinion the same concentration range should have been tested as for RLIT.

Author Response

The manuscript (Novel brassinosteroid analogs with 3,6 dioxo function, 24-Nor-22(S)-hydroxy side chain and p-substituted benzoate function at C-23. Synthesis and evaluation of plant growth effects) deals with the synthesis of new brassinosteroid derivatives and their biological activity. Unfortunately, some important information is missing in the article and should be added. Therefore, I cannot recommend the paper for publication without a major revision.

1) The Results and Discussion section is very poorly written, the discussion is missing and therefore this section should be completely rewritten. (the results of the bioactivity tests are not discussed at all and their in-depth interpretation is also missing).

We have added a new paragraph with a more detailed discussion of our data. It is important to keep on mind that this work is not meant to be a study of biological mechanism of action for BRs analogs. Our main goal is to find a relationship between BRs chemical structure and plant responses. The mechanism by which BRs act in each test is far beyond the subject of this work.

- Tables 2.2 and 2.3 should also include results for negative controls.

Negative controls have been added to both tables

- What was the final concentration of DMSO in the bioassays?

DMSO concentration was 1%, and this has been added in lines 482 and 496

- Why was only 1 concentration (10-8M) tested for the biotest on the second bean internodes, in my opinion the same concentration range should have been tested as for RLIT.

The main reason to use BSI test was to compare the obtained results with those previously reported by Aitken et al., 2024. Additionally, we intend to compare these results with those obtained with RLIT, and therefore we thought that one concentration will be enough.

Round 2

Reviewer 1 Report

Comments and Suggestions for Authors

Accept

Author Response

In this paper, Jorquera et al. synthesized new brassinosteroid analogs obtained from hyodeoxycholic acid, with 3,6 dioxo function, 24-Nor-22(S)-hydroxy side chain and p-substituted benzoate function at C-23. The authors validated this compound in two separate bioassays.

This is a novel research which may benefit many fields, however, the validation of activity of the new compound is insufficient. Additional bioassays or similar bioassays in additional species is required.

The reason why only two different bioassays were used is presented in a more detailed way at the end of the Discussion Section.

Reviewer 2 Report

Comments and Suggestions for Authors

Dear Authors,
 I agree with most of the revisions that you have made, but unfortunately, in my opinion, there is still a lack of discussion in the publication, i.e. a comparison of your results with those of other authors. In my opinion, this should be standard for a journal with an IF of 4.9.
But I leave this assessment to the editor's decision.

Author Response

(The authors gave the same response as above.)
